# Conditional Instrumental Variable Regression with Representation Learning for Causal Inference

**Debo Cheng** [1][*]**, Ziqi Xu** [1,2][*]**, Jiuyong Li** [1]**, Lin Liu** [1]**, Jixue Liu** [1] **& Thuc Duy Le** [1]
University of South Australia [1]     Data61, CSIRO [2]
{firstname.lastname}@unisa.edu.au [1], ziqi.xu@data61.csiro.au [2]

## Abstract

This paper studies the challenging problem of estimating causal effects from observational data, in the presence of unobserved confounders. The two-stage least square (TSLS) method and its variants with a standard instrumental variable (IV) are commonly used to eliminate confounding bias, including the bias caused by unobserved confounders, but they rely on the linearity assumption. Besides, the strict condition of unconfounded instruments posed on a standard IV is too strong to be practical. To address these challenging and practical problems of the standard IV method (linearity assumption and the strict condition), in this paper, we use a conditional IV (CIV) to relax the unconfounded instrument condition of standard IV and propose a non-linear CIV regression with Confounding Balancing Representation Learning, CBRL.CIV, for jointly eliminating the confounding bias from unobserved confounders and balancing the observed confounders, without the linearity assumption. We theoretically demonstrate the soundness of CBRL.CIV. Extensive experiments on synthetic and two real-world datasets show the competitive performance of CBRL.CIV against state-of-the-art IV-based estimators and superiority in dealing with the non-linear situation.

## 1 Introduction

Causal inference is a fundamental problem in various fields such as economics, epidemiology, and medicine (Hernán & Robins, 2010; Imbens & Rubin, 2015). Estimating treatment (i.e., causal) effects of a given treatment on an outcome of interest from observational data plays an important role in causal inference. Nevertheless, estimating causal effects from observational data suffers from the confounding bias caused by confounders[1], and unobserved confounders make causal effect estimation more challenging. When there is an unobserved confounder affecting both the treatment and outcome variables, as illustrated by the causal DAG [2] in Fig. 1 (a), the *endogeneity problem*[3] (Antonakis et al., 2010; Cheng et al., 2024) arises, and in this situation, it is well-known that the causal effect of the treatment on the outcome is non-identifiable (Pearl, 2009; Shpitser & Pearl, 2006).

Instrumental variable (IV) is a powerful approach to addressing the endogeneity problem and identifying causal effects in the presence of unobserved confounders, but under strong assumptions (Anderson & Rubin, 1949; Hernán & Robins, 2006). Typically, the IV approach requires a valid standard IV (denoted as $S$) that satisfies the following conditions: (i) $S$ affects the treatment $W$ (a.k.a. the relevance condition), (ii) the causal effect of $S$ on the outcome $Y$ is only through $W$ (a.k.a. the exclusion condition), and (iii) $S$ and $Y$ are not confounded by measured or unmeasured variables (a.k.a. the unconfounded instrument condition). For instance, the variable $S$ in the causal DAG of Fig. 1 (b) is a standard IV that satisfies the three conditions. Under the linearity assumption, the two-

---

[*]These authors contributed equally to this work.

[1]A confounder is a common cause that affects two variables simultaneously.

[2]A directed acyclic graph (DAG) is a graph that contains only directed edges and no cycles.

[3]The endogeneity comes from the presence of unobserved confounders, which leads to causal ambiguities in the causal relations between the treatment and the outcome (Antonakis et al., 2010; Greenland, 2003).

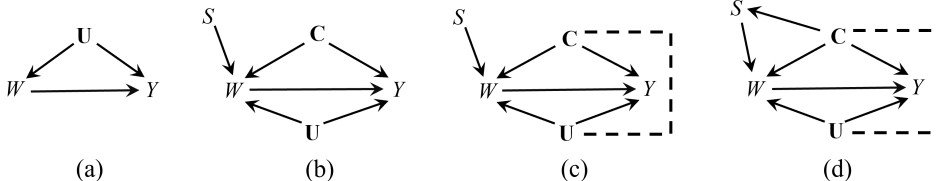

Figure 1: An illustration presents four scenarios: (a) unobserved confounders, (b) standard IV, (c) the imbalance issue with standard IV, and (d) the imbalance issue with CIV. Variables include treatment $W$, outcome $Y$, the set of observed confounders $\mathbf{C}$, the set of unobserved confounders $\mathbf{U}$, and IV $S$. Directed edges indicate causal relationships and dotted lines represent statistical associations. In (a), the causal effect of $W$ on $Y$ is non-identifiable due to $\mathbf{U}$ affecting both $W$ and $Y$; In (b), the causal effect of $W$ on $Y$ can be estimated by using the IV $S$; In (c), the dependence of $\mathbf{U}$ and $\mathbf{C}$ affecting the potential for confounding and distribution imbalance in causal effect estimation, addressed in Wu et al. (2022); In (d), $S$ is a conditional IV (CIV). In this case, $\mathbf{C}$ induces bias across the different groups as specified by the different values of $S$, potentially biasing the estimated causal effect of $W$ on $Y$. This paper addresses this problem.

stage least squares (TSLS) method is often employed to estimate the causal effects in the presence of unobserved confounders using a given standard IV (Angrist & Imbens, 1995; Imbens, 2014).

In many real-world scenarios, non-linear variable relationships combined with the effect of unobserved confounders make estimating causal effects from observational data more challenging. Recently, some methods were proposed to address the challenging case with non-linear relationships when a standard IV is used, such as DeepIV (Hartford et al., 2017), kernelIV regression (Singh et al., 2019), double machine learning IV (DMLIV) method (Syrgkanis et al., 2019), etc. These methods are built on two-stage regression models. Using Fig. 1 (b) as an example, the first stage of such a two-stage approach is to fit a non-linear regression model for the conditional distribution of the treatment $W$ conditioning on the standard IV $S$ and the set of measured pretreatment variables $\mathbf{C}$, i.e., the estimated treatment $\hat{W}$ is obtained based on $P(W \mid S, \mathbf{C})$. The second stage is to fit a non-linear regression model on $Y$ conditioning on $\mathbf{C}$ and $\hat{W}$ obtained in the first stage.

However, due to the spurious associations between $\mathbf{U}$ and $\mathbf{C}$ (as illustrated in Fig. 1 (c)), the distributions of $\mathbf{C}$ in the treated and control groups may be imbalanced (Shalit et al., 2017), which can result in biased estimations in the non-linear outcome regression model (Abadie, 2003), especially when the outcome regression model is misspecified. To address the imbalance issue in the outcome regression model, Wu et al. Wu et al. (2022) proposed a confounder balancing IV regression (CBIV) algorithm, and demonstrated the effectiveness of the confounding balancing representation learning approach. However, CBIV ignores the confounding bias caused by the imbalance distribution of $\mathbf{C}$ across different groups specified by the different values of $S$ in the first stage.

To address the imbalance problems and the practical issues of the standard IV approach (linearity assumption and the strict condition), we consider conditional IV (CIV) in this work. CIV relaxes the third condition of the standard IV, and allows a variable to be a valid IV by conditioning on a set of measured variables (Brito & Pearl, 2002; van der Zander et al., 2015). In real-wolrd applications, a CIV is less restrictive than a standard IV and is easier to find. For example, we aim to estimate the effect of a treatment ($W$) on the recovery from a disease ($Y$). $W$ and $Y$ are affected by unobserved confounders $\mathbf{U}$, such as the socioeconomic status of patients. The variable of clinical practice guidelines ($S$) for treatment decision $W$ is a CIV with the conditioning set ($\mathbf{C}$), which includes factors such as age, health condition, and an individual's propensity to seek medical assistance. This example can be represented by the causal DAG in Fig. 1 (d). In contrast, a standard IV affecting only $W$ but not $Y$ via other paths is difficult to find in the real-world.

A two-stage regression is commonly used to obtain an estimation of the causal effect by an IV or a CIV approach. There are two risks of biases in the estimation using a two-stage solution in Fig. 1 (d). Firstly, the distribution imbalance of $\mathbf{C}$ between the treated and control groups leads to a bias in the outcome regression model. Secondly, unobserved confounders could cause a distribution imbalance of $\mathbf{C}$ between the different groups specified by different values of $S$ as $\mathbf{C}$ leading to a bias in the treatment regression model. Current solutions (Hartford et al., 2017; Wu et al., 2022) consider a standard IV and use a regression model conditioning on $\mathbf{C}$, and they are not applicable to a CIV case since they do not mitigate the confounding bias between $S$ and $W$.

To enhance the practical use of the IV approach, specifically to address the double imbalance problem caused by both observed and unobserved confounders in non-linear cases when using a CIV, we propose CBRL.CIV, a novel Confounding Balancing Representation Learning method for non-linear CIV regression. We further demonstrate its theoretical soundness and practical usefulness. By integrating CIV with confounding balance, CBRL.CIV emerges as a comprehensive solution that adeptly navigates the intricacies of confounder effects in non-linear models, offering a significant advancement in causal inference. We summarise the main contributions of our work as follows.

- We study the problem of causal effect estimation from data with unobserved confounders in a non-linear setting and design a novel and practical CIV method to solve the problem. To the best of our knowledge, this is the first work that considers the imbalance issues in the applications of CIVs when there are finite samples.

- We propose a novel representation learning method, CBRL.CIV for balancing the distributions and removing the confounding bias simultaneously. We theoretically show the soundness of the proposed CBRL.CIV method. Specifically, CBRL.CIV utilises two balance terms to enhance generalisation across different groups specified by different values of $S$ in the treatment network, and different groups specified by different values of $\hat{W}$ in the outcome regression network.

- Extensive experiments are conducted on synthetic and real-world datasets and the results demonstrate the effectiveness of CBRL.CIV in average causal effect estimation.

## 2 PRELIMINARY

We introduce some important notations and definitions used in the paper. Let $W \in \{1, 0\}$ denote a binary treatment variable where $W = 1$ represents treated and $W = 0$ untreated, $Y$ the outcome variable of interest, and $\mathbf{U}$ a set of unobserved confounders. Let $\mathbf{X}$ denote the set of pretreatment variables, namely, all variables in $\mathbf{X}$ are measured before $W$ is measured. Finally, we assume that the complete data are independent and identically distributed realizations of $(W, Y, \mathbf{U}, \mathbf{X})$, and the observed data is $\mathcal{O} = (W, Y, \mathbf{X})$.

We aim at estimating the average causal effect of $W$ on $Y$ from $\mathcal{O}$ with the presence of unobserved confounders $\mathbf{U}$. In the observational data $\mathcal{O}$, for each individual $i$, the potential outcomes are denoted as $Y_i(W = 1)$ and $Y_i(W = 0)$ corresponding to $W$ (Imbens & Rubin, 2015; Rosenbaum & Rubin, 1983). Note that only one of the potential outcomes can be observed for a given individual, which is known as the fundamental problem in causal inference (Imbens & Rubin, 2015).

Furthermore, we use a causal DAG $\mathcal{G} = (\mathbf{V}, \mathbf{E})$ to represent the underlying mechanism of the studied system, where $\mathbf{V}$ represents the set of nodes (variables) of the DAG, and $\mathbf{E} \subseteq \mathbf{V} \times \mathbf{V}$ is the set of directed edges (Pearl, 2009; Spirtes et al., 2000). For any $X_i, X_j \in \mathbf{V}$ and $i \neq j$, the directed edge $X_i \rightarrow X_j$ in the causal DAG $\mathcal{G}$ indicates that $X_i$ is a direct cause (parent) of $X_j$, and $X_j$ is a direct effect (child) of $X_i$. A path from $X_i$ to $X_j$ is a directed path or causal path when all edges along it are directed towards $X_j$. Additionally, on a causal path from $X_i$ to $X_j$, $X_i$ is an ancestor of $X_j$, and conversely, $X_j$ is a descendant of $X_i$. Using a causal DAG, a CIV (Definition 7.4.1 on Page 248 (Pearl, 2009)) can be defined as:

**Definition 1** (Conditional IV). *Let $\mathcal{G} = (\mathbf{V}, \mathbf{E})$ be a causal DAG with $\mathbf{V} = \mathbf{X} \cup \mathbf{U} \cup \{W, Y\}$, a variable $S \in \mathbf{X}$ is a conditional IV w.r.t. $W \rightarrow Y$ if there exists a set of measured variables $\mathbf{C} \subseteq \mathbf{X}$ such that (i) $S \not\perp\!\!\!\perp_d W \mid \mathbf{C}$, (ii) $S \perp\!\!\!\perp_d Y \mid \mathbf{C}$ in $\mathcal{G}_{\underline{W}}$, and (iii) $\forall C \in \mathbf{C}$, $C$ is not a descendant of $Y$, where $\mathcal{G}_{\underline{W}}$ is a manipulated DAG of $\mathcal{G}$ by deleting the direct edge $W \rightarrow Y$ from $\mathcal{G}$, and $\not\perp\!\!\!\perp_d$ and $\perp\!\!\!\perp_d$ are d-connection and d-separation (Pearl, 2009), respectively.*

In the above definition, $\mathbf{C}$ is known as the set of variables that instrumentalise $S$, or the conditioning set of $S$. Note that a CIV is a more general IV than the standard IV and when $\mathbf{C}$ is an empty set in Definition 1, $S$ is a standard IV. More definitions are provided in Appendix A.

# 3 THE PROPOSED CBRL.CIV METHOD

## 3.1 PROBLEM SETTING & ASSUMPTIONS

We assume that a binary CIV $S \in \mathbf{X}$ is known and further assume that the rest variables in $\mathbf{X}$, denoted as $\mathbf{C}$, are pre-IV variables, i.e., they are measured before $S$ is measured. Our problem is illustrated in Fig. 1 (d), a typical setting where a CIV is used, and in the figure, $S$ is a CIV that is instrumentalised by $\mathbf{C}$ (i.e., $\mathbf{C}$ is the conditioning set of $S$). $\mathbf{C}$ affects both $S$ and $W$, and $\mathbf{C}$ affects both $W$ and $Y$. $\mathbf{C}$ has two roles here: observed confounders $\mathbf{C}$ between $W$ and $Y$ and conditioning set of CIV $S$ w.r.t. $W$ and $Y$. As discussed previously, when a CIV is used, in addition to the imbalanced distributions of $\mathbf{C}$ between $W$ and $Y$ as discussed in the previous work (Wu et al., 2022), $\mathbf{C}$ is imbalanced among the groups specified by the different values of $S$. That is, we face a double distribution imbalance problem, one between $S$ and $W$, and one between $W$ and $Y$. Therefore the existing solution addressing the distribution imbalance of $\mathbf{C}$ between $W$ and $Y$ such as CBIV (Wu et al., 2022) cannot be applied because the confounding bias resulted from the imbalance of $\mathbf{C}$ among the groups specified by the different values of $S$ remain unresolved.

To address the double imbalance issue caused in using a CIV approach, especially in non-linear scenarios, we have designed a novel three-stage method based on balanced representation learning techniques (Shalit et al., 2017; Wu et al., 2022).

The following assumptions are essential when using CIV method for causal effect estimation in data.

**Assumption 1.** *(IV positivity) It is almost surely that $0 < P(S = 1 \mid \mathbf{C}) < 1$.*

The IV positivity assumption is necessary for non-parametric identification (Greenland, 2000).

**Assumption 2.** *(Independent compliance type (Cui & Tchetgen Tchetgen, 2021; Hernán & Robins, 2006)) It is almost surely that $\phi(\mathbf{C}) \equiv P(W = 1 \mid S = 1, \mathbf{C}) - P(W = 1 \mid S = 0, \mathbf{C}) = \tilde{\phi}(\mathbf{C}, \mathbf{U})$, where $\tilde{\phi}(\mathbf{C}, \mathbf{U}) \equiv P(W = 1 \mid S = 1, \mathbf{C}, \mathbf{U}) - P(W = 1 \mid S = 0, \mathbf{C}, \mathbf{U})$.*

This assumption means that there is no additive interaction between $S$ on $\mathbf{U}$ in a system for the treatment probability conditioning on $\mathbf{C}$ and $\mathbf{U}$. It also demonstrates that $\mathbf{U}$ is independent of an individual's compliance type (Imbens & Rubin, 2015). In a binary $S$, there are four compliance types for an individual: *Compliers*, *Always-takers*, *Never-takers* and *Defiers* (Angrist & Imbens, 1995; Imbens & Rubin, 2015). The details of compliance types are provided in Appendix A.

## 3.2 CBRL.CIV

Our proposed CBRL.CIV method tackles the double distribution imbalance issue by learning a confounding balancing representation of $\mathbf{C}$, denoted as $\mathbf{Z}$ using a representation network $\psi_\theta(\cdot)$, i.e., $\mathbf{Z} = \psi_\theta(\mathbf{C})$ to achieve confounding balance in counterfactual inference and remove the confounding bias from unobserved confounders without linearity assumption. CBRL.CIV consists of three components, CIV regression for balancing the distributions of $\mathbf{C}$ between $S$ and $W$, treatment regression for balancing the distribution between $W$ and $Y$, and outcome regression using the learned balancing representation $\mathbf{Z}$.

**CIV Regression for Confounding Balance.** To balance the distributions of $\mathbf{C}$ between different values of the CIV $S$, we need to ensure that $S$ and the representation $\mathbf{Z}$ are independent conditioning on $\mathbf{C}$. It is challenging to directly learn a representation $\mathbf{Z}$ such that $S \perp\!\!\!\perp \mathbf{Z} \mid \mathbf{C}$ from the data. We propose to achieve this in two steps. In Step one, we employ a logistic regression network $\varphi_\mu(\mathbf{c}_i)$ with $\mu$ as a parameter of the network for an individual $i$ to predict the conditional probability distribution of the CIV $S$, $P(S \mid \mathbf{C})$, and we sample $\hat{S}$ from $P(S \mid \mathbf{C})$, i.e., $\hat{S} \sim P(S \mid \mathbf{C})$. For the CIV regression model, we have the following loss function:

$$L_S = \frac{1}{n} \sum_{i=1}^{n} (s_i \log(\varphi_\mu(\mathbf{c}_i)) + (1 - s_i)(1 - \log(\varphi_\mu(\mathbf{c}_i)))) \tag{1}$$

where $n$ is the number of samples in the data. Step two learns $\mathbf{Z}$ such that $\hat{S} \perp\!\!\!\perp \mathbf{Z}$. For achieving this goal, we train the representation network $\mathbf{Z} = \psi_\theta(\mathbf{C})$ with the learnable parameter $\theta$ and minimise the discrepancy of the distributions of $\mathbf{Z}$ for different values of $\hat{S}$ by using $IPM(\cdot)$, which is the

integral probability metric for measuring the discrepancy of distributions. The above requirement for confounding balance is formally stated as the following objective:

$$D(\hat{S}, \psi_\theta(\mathbf{C})) = IPM(\{\psi_\theta(\mathbf{C})P(\hat{s}_i = 0 \mid \mathbf{c}_i)\}_{i=1}^n, \{\psi_\theta(\mathbf{C})P(\hat{s}_i = 1 \mid \mathbf{c}_i)\}_{i=1}^n) \quad (2)$$

where $\{\psi_\theta(\mathbf{C})P(\hat{s}_i = s \mid \mathbf{c}_i)\}_{i=1}^n, s \in \{0, 1\}$ is the distribution of the representation $\mathbf{Z}$ (i.e., $\psi_\theta(\mathbf{C})$) in the sub-group $\hat{S} = s$. In this way, we obtain $\hat{S} \perp\!\!\!\perp \mathbf{Z}$, which implies $S \perp\!\!\!\perp \mathbf{Z} \mid \mathbf{C}$ because $\hat{S}$ is obtianed by conditioning on $\mathbf{C}$ and $\mathbf{Z}$ is also obtained by conditioning on $\mathbf{C}$. We utilise the Wasserstein distance as the metric for evaluating the discrepancies for CFR, CBIV, and our method CBRL.CIV, enabling a fair comparison in our experiments because Wasserstein distance is used in the implementations of CFR and CBIV.

**Treatment Regression with Confounding Balance.** To address the distribution imbalance of $\mathbf{C}$ between $W$ and $Y$ and the effect of unobserved confounders $U$, we follow the previous non-linear IV-based methods, such as DeepIV (Hartford et al., 2017) and CBIV (Wu et al., 2022), to build a treatment regression network by using the estimated $\hat{S}$ obtained in the first stage and $\mathbf{C}$ to regress $W$ for obtaining an estimated $\hat{W}$ that is not affected by $\mathbf{U}$. Specifically, we use a logistic regression network $\varphi_\nu(\hat{s}_i, \mathbf{c}_i)$ with the learnable parameter $\nu$ to calculate the conditional probability distribution of the treatment $W$, i.e., $\hat{W} \sim P(W \mid \hat{S}, \mathbf{C})$, and optimise the following loss function:

$$L_W = \frac{1}{n} \sum_{i=1}^n (w_i \log(\varphi_\nu(\hat{s}_i, \mathbf{c}_i)) + (1 - w_i)(1 - \log(\varphi_\nu(\hat{s}_i, \mathbf{c}_i)))) \quad (3)$$

After obtaining $\hat{W}$, to achieve $\hat{W} \perp\!\!\!\perp \mathbf{Z}$ for alleviating the effect of the distribution imbalance of $\mathbf{C}$ between $W$ and $Y$, we minimise the discrepancy of the distributions $\mathbf{Z}$ for different values of $\hat{W}$, and the objective function can be formed as:

$$D(\hat{W}, \psi_\theta(\mathbf{C})) = IPM(\{\psi_\theta(\mathbf{C})P(\hat{w}_i = 0 \mid \hat{s}_i, \mathbf{c}_i)\}_{i=1}^n, \{\psi_\theta(\mathbf{C})P(\hat{w}_i = 1 \mid \hat{s}_i, \mathbf{c}_i)\}_{i=1}^n) \quad (4)$$

where $\{\psi_\theta(\mathbf{C})P(\hat{w}_i = w \mid \hat{s}_i, \mathbf{c}_i)\}_{i=1}^n, w \in \{0, 1\}$ represents the distribution of the representation $\mathbf{Z} = \psi_\theta(\mathbf{C})$ in the sub-group $\hat{W} = w$. The confounding balance term is a constraint term that is included in the learning process to ensure that $\psi_\theta(\mathbf{C})$ and $\hat{W}$ are independent, i.e., $\hat{W} \perp\!\!\!\perp \mathbf{Z}$.

**Outcome Regression.** In the final stage, we propose to regress the outcome using $\mathbf{Z}$ (i.e., $\psi_\theta(\mathbf{C})$) and the estimated $\hat{W}$ obtained in the treatment regression stage. Furthermore, we use two separate regression networks for counterfactual inference, as done in Hartford et al. (2017). Both networks, denoted as $f_{\gamma^1}(\psi_\theta(\mathbf{C}))$ and $f_{\gamma^0}(\psi_\theta(\mathbf{C}))$, have two learnable parameters $\gamma^1$ and $\gamma^0$, respectively.

The estimated treatment $\hat{W} \sim P(W \mid \hat{S}, \mathbf{C})$ and the balancing representation $\psi_\theta(\mathbf{C})$ are used to regress $Y$ in the outcome regression network. The objective function for the two separate regression networks is defined as follows:

$$L_Y = \frac{1}{n} \sum_{i=1}^n \left( y_i - \sum_{w_i \in \{0,1\}} f_{\gamma^{w_i}}(\psi_\theta(\mathbf{C}))P(w_i \mid \hat{s}_i, \mathbf{c}_i) \right)^2 \quad (5)$$

where $y_i$ is the value of observed outcome, and $P(w_i \mid \hat{s}_i, \mathbf{c}_i)$ is obtained from the treatment regression network $\varphi_\nu(\hat{s}_i, \mathbf{c}_i)$ in the second stage.

**Optimisation.** We employ a three-stage optimisation for our proposed CBRL.CIV method. Firstly, we optimise the CIV regression network $\varphi_\mu(\mathbf{c}_i)$ for minimising the loss function $L_S$ in Eq. 1. Then, we optimise the treatment regression network $\varphi_\nu(\hat{s}_i, \mathbf{c}_i)$ for minimising the loss function $L_W$ in Eq. 3. Finally, we simultaneously optimise the two confounding balance representation learning networks and the outcome regression network by adding the two confounding balance terms as a type of regularisation to the objective function of the outcome regression network, as follows:

$$min_{\theta, \gamma^0, \gamma^1} L_Y + \alpha D(\hat{W}, \psi_\theta(\mathbf{C})) + \beta D(\hat{S}, \psi_\theta(\mathbf{C})) \quad (6)$$

where $\alpha$ and $\beta$ are tuning parameters.

After we obtain $\psi_\theta(\mathbf{C})$ and parameters $\gamma^0$ and $\gamma^1$, the average causal effect (ACE) of $W$ on $Y$ can be calculated by

$$\hat{ACE} = \mathbb{E}[f_{\gamma^1}(\psi_\theta(\mathbf{C})) - f_{\gamma^0}(\psi_\theta(\mathbf{C}))] \quad (7)$$

The pseudo-code of the proposed CBRL.CIV method is provided in Appendix B.1.

### 3.3 THEORETICAL ANALYSIS

The deep learning based IV estimator, DeepIV (Hartford et al., 2017), provides the counterfactual prediction function:

$$f(W, \mathbf{C}) \equiv g(W, \mathbf{C}) + \mathbb{E}[\mathbf{U} \mid \mathbf{C}] \tag{8}$$

where $\mathbf{U}$ is only conditioning on $\mathbf{C}$, but not $W$. $\mathbb{E}[\mathbf{U} \mid \mathbf{C}]$ is non-zero, but stays constant with the change of $W = 1$ to $W = 0$. With a valid standard IV $S$, the counterfactual prediction function $f(W, \mathbf{C})$ can be solved over the observed confounders:

$$\mathbb{E}[Y \mid S, \mathbf{C}] = \int f(W, \mathbf{C}) dP(W \mid S, \mathbf{C}) \tag{9}$$

where $dP(W \mid S, \mathbf{C})$ is the conditional treatment distribution. The relationship in Eq.(9) defines an *inverse problem* for the function $f$ by using two observed functions $\mathbb{E}[Y \mid S, \mathbf{C}]$ and $P(W \mid S, \mathbf{C})$ (Hartford et al., 2017; Newey & Powell, 2003). Most IV methods (Chen & Pouzo, 2012; Singh et al., 2019) employ a two-stage process for solving this problem: estimating $\hat{P}(W \mid S, \mathbf{C})$ for the true $P(W \mid S, \mathbf{C})$, and then approximating $f(W, \mathbf{C})$ given the estimated $\hat{P}(W \mid S, \mathbf{C})$.

We have the following theorem to identify (solve) the counterfactual prediction in Eq.(9) by using a CIV and deep neural networks Bennett et al. (2019); LeCun et al. (2015) in our problem setting.

**Theorem 1.** *Given the observational data $\mathcal{O} = (\mathbf{C}, S, W, Y)$ generated from a causal DAG $\mathcal{G} = (\mathbf{C} \cup \mathbf{U} \cup \{S, W, Y\}, \mathbf{E})$ where $\mathbf{E}$ is the set of directed edges as shown in Fig. 1 (d), suppose that $S$ is a CIV and $\mathbf{C}$ is a set of pre-IV variables. Under Assumptions 1 and 2, if the learned representation $\mathbf{Z} = \psi_\theta(\mathbf{C})$ of $\mathbf{C}$ is independent of estimated $\hat{S}$ and $\hat{W}$, then $f(W, \mathbf{C})$ can be identified by using the $\hat{S}$ and the representation $\mathbf{Z}$ as: $\mathbb{E}[Y \mid S, \mathbf{C}] = \int f(W, \mathbf{Z}) dP(W \mid \hat{S}, \mathbf{C})$, where $P(W \mid \hat{S}, \mathbf{C})$ is the estimated conditional treatment distribution by using $\hat{S}$ and $\mathbf{C}$.*

The proof is given in Appendix B.2. Theorem 1 ensures the soundness of the proposed CBRL.CIV method by solving an inverse problem for counterfactual outcomes prediction.

## 4 EXPERIMENTS

The goal of the experiments is to assess the performance of the proposed CBRL.CIV method in ACE estimation from data in the presence of unobserved confounders. Thus, we conduct extensive experiments on synthetic datasets since they allow the true ACE to be known. To show the potential of the proposed CBRL.CIV method in real-world applications, we also evaluate the performance of CBRL.CIV on two real-world datasets that have been widely used in the evaluation of IV-based estimators (Abadie, 2003; Card, 1993; Cheng et al., 2024). Due to the page limitation, we report the ablation study, and the effect of $\alpha$ and $\beta$ on CBRL.CIV in Appendix C.3 and C.4, respectively.

### 4.1 EXPERIMENTAL SETUP

**Baselines.** We assess the performance of CBRL.CIV by comparing it with eight state-of-the-art causal effect estimators for causal effect estimation from observational data. These estimators can be categorised into three groups: (I) most closely related work: CBIV, the confounding balancing IV regression Wu et al. (2022); (II) Other IV methods using machine learning/representation learning, including (1) the DeepIV (Hartford et al., 2017) (a deep neural network based IV estimator), (2) OrthIV Sjolander & Martinussen (2019) (an orthogonal regularisation machine learning based IV estimator), (3) DMLIV (Syrgkanis et al., 2019) (double machine learning based IV estimator), (4) DRIV (Syrgkanis et al., 2019) (doubly robust learning based IV estimator), (5) SDRIV (Chernozhukov et al., 2018) (sparse linear doubly DRIV), and (6) ForestDRIV (Syrgkanis et al., 2019) (random forest DRIV); (III) Other estimator, not IV based, but a popular representation learning method which considers confounding balance, i.e., CounterFactual Regression by using deep representation learning (CFR) (Shalit et al., 2017).

**Implementation.** We have implemented CBRL.CIV using Python programming language. We used multi-layer perceptrons with ReLU activation function and BatchNorm to implement the logistic regression networks for CIV regression and treatment regression. For optimising the CIV

Table 1: The table summarises the estimation error (mean±STD) over 30 synthetic datasets with different settings. The smallest estimation error in each setting is highlighted. Note that CBRL.CIV obtains the smallest estimation error in terms of mean and STD.

| Methods | Within-Sample | | | Out-of-Sample | | |
|---|---|---|---|---|---|---|
| | Syn-4-4 | Syn-8-8 | Syn-12-12 | Syn-4-4 | Syn-8-8 | Syn-12-12 |
| DeepIV | 1.58±0.11 | 1.49±0.14 | 1.45±0.17 | 1.58±0.17 | 1.45±0.18 | 1.45±0.22 |
| OrthIV | 0.69±0.43 | 1.86±0.15 | 1.72±0.08 | 3.86±5.42 | 2.1±0.29 | 1.86±0.11 |
| DMLIV | 0.69±0.45 | 1.84±0.14 | 1.73±0.08 | 2.06±1.66 | 2.05±0.32 | 1.86±0.12 |
| DRIV | 1.96±0.51 | 1.14±0.54 | 0.6±0.35 | 2.03±0.34 | 1.42±0.39 | 0.92±0.43 |
| SDRIV | 1.91±0.36 | 0.72±0.5 | 0.46±0.35 | 2.13±0.39 | 1.09±0.44 | 0.52±0.49 |
| ForestDRIV | 1.87±0.34 | 0.72±0.4 | 0.46±0.36 | 1.99±0.41 | 0.94±0.6 | 0.75±0.52 |
| CFR | 0.6±0.38 | 0.58±0.3 | 0.51±0.24 | 0.6±0.38 | 0.58±0.31 | 0.51±0.25 |
| CBIV | 0.48±0.27 | 0.4±0.19 | 0.41±0.14 | 0.48±0.27 | 0.4±0.19 | 0.41±0.14 |
| CBRL.CIV | **0.14±0.05** | **0.14±0.03** | **0.12±0.03** | **0.14±0.05** | **0.14±0.03** | **0.12±0.03** |

and treatment regression networks, we used stochastic gradient descent (SGD) (Duchi et al., 2011). Furthermore, we used the ADAM method (Kingma & Ba, 2014) to optimise the outcome network jointing the confounding balance modules. To prevent overfitting, we employed an $l_2$-regularisation term to regularise the objective function in Eq. (6). The code of CBRL.CIV, network parameters, and parameter tuning are provided in the supplementary materials. The implementations and parameter settings of compared estimators are provided in Appendix C.1.

**Evaluation Metrics.** For the experiments on synthetic and two real-world datasets, we use the estimated ACE error, $\left|(\hat{ACE} - ACE)\right|$ to evaluate the performance of all estimators in ACE estimation from data with unobserved confounders.

### 4.2 SIMULATION STUDY

We follow the data generation process in (Hassanpour & Greiner, 2020; Wu et al., 2022) to generate synthetic datasets for the evaluation. Note that the underlying causal DAG in Fig. 1 (d) is used to generate all synthetic datasets for evaluation of the performance of CBRL.CIV is relative to the double imbalance issues in causal effect estimation from data with unobserved confounders. The details of the synthetic data generation process and more results are provided in Appendix C.2.

We assess the performance of CBRL.CIV under different settings of the number of observed and unobserved confounders, and we denote the settings as 'Syn-p-q', i.e., a setting/dataset with $p$ observed confounders and $q$ unobserved confounders. In our experiments, we sample 10k individuals with three settings, Syn-4-4, Syn-8-8, and Syn-12-12. We report the mean and the standard deviation (STD) of the estimation error of all methods in Table 1. Note that 'Within-Sample' denotes the estimation error calculated over the training samples, and 'Out-of-Sample' implies the estimation error calculated over the testing samples.

From the estimation errors of all estimators in Table 1, we have four conclusions: (1) The proposed CBRL.CIV method obtains the smallest estimation error on all synthetic datasets because of CBRL.CIV balances the distribution of $\mathbf{C}$ between $\hat{W}$ and $Y$ in the outcome regression stage, as well as the distribution of $\mathbf{C}$ between $\hat{S}$ and $W$ in the treatment regression. (2) CBIV has the second smallest estimation error but is worse than CBRL.CIV since it ignores the distribution imbalance of $\mathbf{C}$ between $\hat{S}$ and $W$ in the treatment regression. (3) The estimators in the second group have a large estimation error since these methods ignore the confounding bias in both stages. (4) The confounding balance-based estimator, CFR has the third smallest estimation error, but CFR does not deal with the affect of $\mathbf{U}$.

We also evaluate the impact of sample size on the performance of CBRL.CIV uses synthetic datasets with a range of sample sizes: $2k, 4k, 6k, 8k, 10k$, and $20k$. The estimation errors of all estimators on Out-of-Sample are visualised with boxplots in Fig. 3. From Fig. 3, we have the following observations: (1) CBRL.CIV has the lowest error and smallest variance of error in different sample sizes. (2) As the sample size increases, the variance of the estimation error of CBRL.CIV decreases slightly. (3) CBRL.CIV has the narrowest upper and lower bounds compared to all other methods.

In summary, CBRL.CIV achieves the best performance among all estimators in ACE estimation from synthetic data in the presence of unobserved confounders.

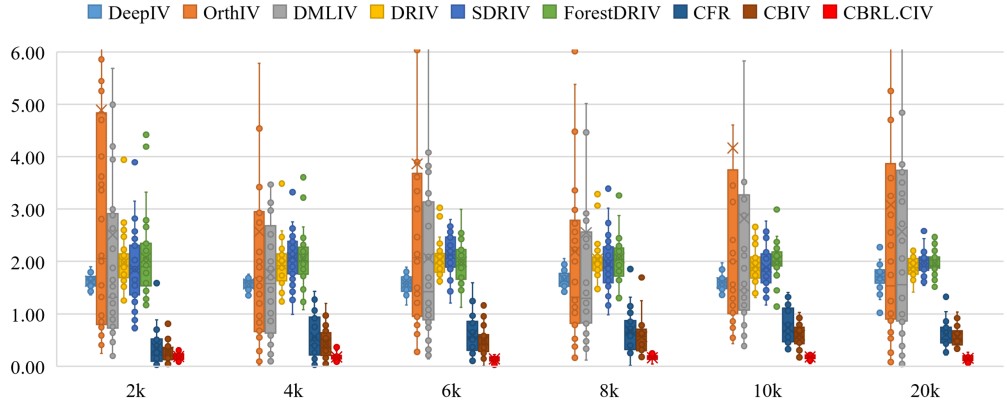

Figure 2: The experimental results of all estimators on synthetic datasets with unobserved confounders in non-linear scenarios. Our CBRL.CIV achieves the lowest error in ACE estimation. And as the sample increases, CBRL.CIV has a slight decrease in variance slightly.

Table 2: The estimation errors (mean±STD) over the out of samples on two real-world datasets with different dimensions of $p$ and $q$.

| Methods | IHDP-2-2 | IHDP-4-4 | IHDP-6-6 | Twins-2-2 | Twins-4-4 | Twins-6-6 |
|---|---|---|---|---|---|---|
| DeepIV | 2.642±0.407 | 2.892±0.262 | 2.471±0.419 | 0.088±0.045 | 0.081±0.070 | 0.081±0.050 |
| OrthIV | 14.678±64.451 | 3.702±3.843 | 11.571±17.026 | 0.465±0.379 | 0.533±0.336 | 0.675±0.882 |
| DMLIV | 7.472±10.153 | 8.945±9.029 | 4.805±2.629 | 3.167±8.197 | 1.695±1.839 | 0.412±0.269 |
| DRIV | 3.865±1.501 | 7.163±9.231 | 4.875±3.889 | 0.588±0.534 | 0.602±0.853 | 0.589±0.421 |
| SDRIV | 7.373±17.763 | 4.009±6.502 | 6.373±7.868 | 0.799±0.585 | 0.591±0.403 | 1.773±4.778 |
| ForestDRIV | 16.252±29.596 | 12.197±35.402 | 5.283±5.484 | 0.763±0.647 | 0.357±0.307 | 0.673±0.544 |
| CFR | 2.570±1.064 | 1.754±0.501 | 2.917±0.419 | 0.037±0.008 | 0.039±0.021 | 0.040±0.026 |
| CBIV | 2.312±0.934 | 1.349±0.452 | 2.846±0.490 | 0.033±0.011 | 0.034±0.026 | 0.035±0.016 |
| **CBRL.CIV** | **0.251±0.082** | **0.454±0.068** | **0.108±0.063** | **0.030±0.019** | **0.027±0.018** | **0.030±0.024** |

## 4.3 EXPERIMENTS ON TWO REAL-WORLD DATASETS

We also conduct experiments on two real-world datasets, IHDP (Hill, 2011) and Twins (Shalit et al., 2017), to evaluate the performance of CBRL.CIV method and demonstrate its potential in real-world applications. The two datasets are widely used in the evaluation of data-driven causal effect estimators (Cheng et al., 2024; Guo et al., 2020).

**IHDP:** The Infant Health and Development Program (IHDP) dataset consists of 747 infants (139 treated infants and 608 control infants). A total of $p + q$ variables, serving as confounders, are selected from the original covariates that are ancestors of, or direct causes of $Y$, in which $p$ variables are observed confounders $\mathbf{C}$, and $q$ variables are unobserved confounders $\mathbf{U}$ (where $p$ and $q$ take values of 2, 4, and 6 respectively). Then we have the CIV assignment probability: $P(S \mid \mathbf{C}) = \frac{1}{1+exp(-(\sum_{i=1}^{p} C_i + \sum_{i=1}^{q} U_i))}$, $S \sim Bern(P(S \mid \mathbf{C}))$, and the treatment assignment probability: $P(W \mid S, \mathbf{C}) = \frac{1}{1+exp(-(S*\sum_{i=1}^{p} C_i + \sum_{i}^{p} C_i + \sum_{i=1}^{q} U_i))}$, $W \sim Bern(P(W \mid S, \mathbf{C}))$, where $Bern(\cdot)$ is the Bernoulli distribution.

**Twins:** The dataset is collected from the recorded twin births and deaths in the USA between 1989-1991 (Almond et al., 2005). Similar to Wu et al. Wu et al. (2022), we use 5,271 records from same-sex twins with weights less than 2,000g from the original data. Following the same process with IHDP, we choose $p + q$ variables from the original covariates as confounders and generate the CIV $S$ and the treatment $W$.

The experiments on the two real-world datasets are also conducted over 30 replications with the 63/27/10 splitting of train/validation/test samples. Following the work in (Wu et al., 2022), in each replication, we shuffle the data and split the data into train/validation/test samples randomly.

**Results.** Table 2 shows the results where IHDP-$p$-$q$ or Twins-$p$-$q$, e.g. IHDP-2-4 denotes the use of a dataset with 2 observed confounders and 4 unobserved confounders. From Table 2, we have

the following observations: (1) CBRL.CIV achieves the lowest estimation errors in all experimental settings of both datasets. (2) CBIV achieves the second-best performance on both datasets. (3) CFR and DeepIV outperform the traditional IV methods.

In a word, our CBRL.CIV method obtains the best performance on both synthetic and real-world datasets compared to the state-of-the-art IV methods and confounding balance based methods.

## 5 RELATED WORKS

The IV approach is a useful way to eliminate the confounding bias due to unobserved confounders in causal effect estimation (Anderson & Rubin, 1949; Cheng et al., 2023; Wang et al., 2022). Two-Stage Least Squares (TSLS) is one of the most commonly used IV-based methods for causal effect estimation from data in the presence of unobserved confounders through linear regression (Angrist & Imbens, 1995; Imbens, 2014). In non-linear setting, some methods use non-linear regression to regress the treatment conditioning on the IV and the other observed confounders in the first stage, and then regress the outcome variable by using the resampled treatment and the observed confounders (Newey & Powell, 2003). Recently, machine learning based IV methods have been developed for reducing estimation bias, such as Kernel IV (Singh et al., 2019), OrthIV (Sjolander & Martinussen, 2019), and Deep Features IV Regress (Xu et al., 2021). Furthermore, DeepIV Hartford et al. (2017) and OneSIV (Lin et al., 2019) fit the treatment assignment based on the IV and the observed confounders, and then resample the treatment based on the fitted treatment assignment mechanism. CBIV (Wu et al., 2022) combines confounding balancing with IV regression to remove the confounding bias between $W$ and $Y$ caused by the unobserved confounders and the observed confounders. Different from these methods, our CBLR.CIV addresses the confounding bias between $S$ and $W$, as well as the confounding bias between $W$ and $Y$. Therefore, CBLR.CIV has a wider application than a method based on a standard IV and it also deals with non-linear cases.

Another line of related works is confounding balance with representation learning (Hassanpour & Greiner, 2020; Shalit et al., 2017). They consider confounding balancing as a domain adaption problem (Daumé III, 2007) under the unconfoundedness assumption (Imbens & Rubin, 2015). In contrast, our CBLR.CIV does not rely on the unconfoundedness, and it reduces the confounding bias due to both unobserved and observed confounders by using representation learning techniques.

## 6 CONCLUSION

In this paper, we propose a novel method called Confounding Balancing Representation Learning for non-linear CIV regression (CBRL.CIV) for causal effect estimation from data with unobserved confounders in non-linear scenarios. CBRL.CIV addresses the imbalance problems of observed confounders and the confounding bias of unobserved confounders by using a balancing representation learning network to obtain a balancing representation of the observed confounders and removing the confounding bias caused by unobserved confounders simultaneously in a non-linear system. Extensive experiments conducted on synthetic and two real-world datasets have demonstrated the effectiveness of CBRL.CIV in average causal effect estimation from data with unobserved confounders. The proposed CBRL.CIV method makes significant contributions to causal effect estimation by considering the distribution imbalance of observed confounders between the CIV and the treatment, and the imbalance distributions of observed confounders between the treatment and the outcome simultaneously, which is the first work to address the challenging problem in causal inference. The proposed CBRL.CIV method has potential applications in various fields, including healthcare, social sciences, and economics, where causal effect estimation is essential for decision-making. The CBRL.CIV method relies on some specific assumptions, such as the existence of a valid CIV and the assumption on independent compliance type (Assumption 2). If these assumptions fail, the method may produce incorrect causal effects. It is vital to verify these conditions and conduct robustness checks and sensitivity analyses to ensure the validity of the causal estimates by CBRL.CIV when the domain experts are not sure whether the conditions are satisfied.

ACKNOWLEDGMENTS

We wish to acknowledge the support from the Australian Research Council Discovery Project 230101122. Thuc Duy Le is supported by DECRA (DE200100200).

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

## A   APPENDIX

This is the appendix to the paper 'Conditional Instrumental Variable Regression with Representation Learning for Causal Inference'. We provide additional definitions of causal DAG, the pseudo-code of CBRL.CIV, the proof of Theorem 1, the details of synthetic and semi-synthetic real-world datasets, and more experimental results.

## A   PRELIMINARY

The definitions of Markov property and faithfulness are usually required in causal graphical modelling.

**Definition 2** (Markov property Pearl (2009)). *Given a DAG $\mathcal{G} = (\mathbf{X}, \mathbf{E})$ and the joint probability distribution of $\mathbf{X}$ ($P(\mathbf{X})$), $\mathcal{G}$ satisfies the Markov property if for $\forall X_i \in \mathbf{X}$, $X_i$ is probabilistically independent of all of its non-descendants, given the set of parent nodes of $X_i$.*

**Definition 3** (Faithfulness Spirtes et al. (2000)). *A DAG $\mathcal{G} = (\mathbf{X}, \mathbf{E})$ is faithful to a joint distribution $P(\mathbf{X})$ over the set of variables $\mathbf{X}$ if and only if every independence present in $P(\mathbf{X})$ is entailed by $\mathcal{G}$ and satisfies the Markov property. A joint distribution $P(\mathbf{X})$ over the set of variables $\mathbf{X}$ is faithful to the DAG $\mathcal{G}$ if and only if the DAG $\mathcal{G}$ is faithful to the joint distribution $P(\mathbf{X})$.*

When the faithfulness assumption is satisfied between a joint distribution $P(\mathbf{X})$ and a DAG of a set of variables $\mathbf{X}$, the dependency/independency relations among the variables can be read from the DAG Pearl (2009); Spirtes et al. (2000). In a DAG, d-separation is a well-known graphical criterion used to read off the conditional independence between variables entailed in the DAG when the Markov property and faithfulness are satisfied Pearl (2009); Spirtes et al. (2000).

**Definition 4** (d-separation Pearl (2009)). *A path $\pi$ in a DAG $\mathcal{G} = (\mathbf{X}, \mathbf{E})$ is said to be d-separated (or blocked) by a set of nodes $\mathbf{M}$ if and only if (i) $\pi$ contains a chain $X_i \rightarrow X_k \rightarrow X_j$ or a fork $X_i \leftarrow X_k \rightarrow X_j$ such that the middle node $X_k$ is in $\mathbf{M}$, or (ii) $\pi$ contains a collider $X_k$ such that $X_k$ is not in $\mathbf{M}$ and no descendant of $X_k$ is in $\mathbf{M}$. A set $\mathbf{M}$ is said to d-separate $X_i$ from $X_j$ ($X_i \perp\!\!\!\perp X_j \mid \mathbf{M}$) if and only if $\mathbf{M}$ blocks every path between $X_i$ to $X_j$. Otherwise, they are said to be d-connected by $\mathbf{M}$, denoted as $X_i \not\perp\!\!\!\perp X_j \mid \mathbf{M}$.*

The instrumental variable (IV) approach is a powerful tool for estimating causal effects from observational data in the presence of unobserved confounders and noncompliance with the assigned treatment $W$ Angrist & Imbens (1995); Imbens (2014); Imbens & Rubin (2015). The main strata are based on the instrument assignment $S$ to follow the treatment $W$ (or not). *Compliers* always comply with $W$: $W(S = 1) = 1$ and $W(S = 0) = 0$; *Always-takers* always follow $W$: $W(S = 1) = W(S = 0) = 1$; *Never-takers* never follow $W$: $W(S = 1) = W(S = 0) = 0$; *Defiers* always against $W$: $W(S = 1) = 0$ and $W(S = 0) = 1$.

# B   THE PROPOSED CBRL.CIV METHOD

In this section, the pseudo-code of CBRL.CIV, the hyperparameters, the tuning parameters setting, and the proof of Theorem 1 are provided.

## B.1   PSEUDO-CODE OF CBRL.CIV METHOD

The proposed three networks can be optimized sequentially. The loss functions of the three networks are formulated as follows.

$$min_\mu L_S = \frac{1}{n} \sum_{i=1}^{n} (s_i log(\varphi_\mu(\mathbf{c}_i)) + (1 - s_i)(1 - log(\varphi_\mu(\mathbf{c}_i)))) \tag{10}$$

$$min_\nu L_W = \frac{1}{n} \sum_{i=1}^{n} (w_i log(\varphi_\nu(\hat{s}_i, \mathbf{c}_i)) + (1 - w_i)(1 - log(\varphi_\nu(\hat{s}_i, \mathbf{c}_i)))) \tag{11}$$

$$min_{\theta,\gamma^0,\gamma^1} L_Y + \alpha L_{CBW} + \beta L_{CBS}$$

$$= \frac{1}{n} \sum_{i=1}^{n} \left( y_i - \sum_{\hat{w}_i \in \{0,1\}} f_{\gamma^{\hat{w}_i}}(\psi_\theta(\mathbf{C})) P(w_i \mid \hat{s}_i, \mathbf{c}_i) \right)^2 \tag{12}$$

$$+ \alpha D(\hat{W}, \psi_\theta(\mathbf{C})) + \beta D(\hat{S}, \psi_\theta(\mathbf{C}))$$

where $\alpha$ and $\beta$ are tuning parameters. $L_{CBW}$ and $L_{CBS}$ are the confounding balance constraints in Eq.s (2) and (4), respectively. The pseudo-code of CBRL.CIV is listed in Algorithm 1.

**Network Structures & Tuning Parameters.**   For the CIV and treatment regression networks, we use multi-layer perceptrons with ReLU activation functions for the logistic regression networks $\varphi_\mu$ and $\varphi_\nu$ respectively. We optimize the loss functions $L_S$ and $L_W$ using SGD with a learning rate of 0.05. For the outcome regression network, we use ADAM with a learning rate of 0.0005 to optimize the loss function $L_Y + \alpha L_{CBW} + \beta L_{CBS}$. For the two tuning parameters $\alpha$ and $\beta$, we select them from the range $\{0.0001, 0.001, 0.01, 0.1, 1\}$.

## B.2   THEORETICAL ANALYSIS

**Theorem 2.** *Given the observational data $\mathcal{O} = (\mathbf{C}, S, W, Y)$ generated from a causal DAG $\mathcal{G} = (\mathbf{C} \cup \mathbf{U} \cup \{S, W, Y\}, \mathbf{E})$ where $\mathbf{E}$ is the set of directed edges as shown in Fig. 1 (d) in the main text, suppose that $S$ is a CIV and $\mathbf{C}$ is a set of pre-CIV variables. Under Assumptions 1 and 2, if the learned representation $\mathbf{Z} = \psi_\theta(\mathbf{C})$ of $\mathbf{C}$ is independent of the estimated $\hat{S}$ and $\hat{W}$, then $f(W, \mathbf{C})$ can be identified by using $\hat{S}$ and $\mathbf{Z}$ by solving the inverse problem using $\hat{S}$ and $\mathbf{Z}$ as: $\mathbb{E}[Y \mid S, \mathbf{C}] = \int f(W, \mathbf{Z}) dP(W \mid \hat{S}, \mathbf{C})$, where $P(W \mid \hat{S}, \mathbf{C})$ is the estimated conditional treatment distribution by using $\hat{S}$ and $\mathbf{C}$.*

*Proof.* We prove that the estimated $\hat{S}$, $\hat{W}$, and the learned representation $\mathbf{Z}$ by CBRL.CIV can be used to identify $f(W, \mathbf{C})$ from the observational data $\mathcal{O}$. We first note that $S$ and $W$, as well as $S$ and $Y$ are confounded by $\mathbf{C}$ respectively in $\mathcal{O}$ and the CBRL.CIV method uses the previously described three-stage strategy to adjust for the confounding bias due to $\mathbf{C}$ and $\mathbf{U}$.

In the CIV regression stage, we aim to obtain $S \perp\!\!\!\perp \mathbf{Z} \mid \mathbf{C}$ to eliminate the confounding bias between $S$ and $W$. For this, we first perform logistic regression using deep neural networks for predicting the conditional probability distribution of the CIV $S$ conditioning on $\mathbf{C}$, namely $P(S \mid \mathbf{C})$. Next, $\hat{S}$ is sampled from $P(S \mid \mathbf{C})$ assuming $P(S \mid \mathbf{C}) = P(S \mid \mathbf{C}, \mathbf{Z})$ since then in the confounding balance phase, the learned representation $\mathbf{Z}$ and the estimated $\hat{S}$ are forced to be independent, i.e., $\hat{S} \perp\!\!\!\perp \mathbf{Z}$ when the optimal solution of Eq. (6) in the main text is obtained. Therefore, the causal relationship between $\hat{S}$ and $W$ is not confounded by $\mathbf{Z}$.

In the treatment regression stage, we aim to obtain $W \perp\!\!\!\perp \mathbf{Z} \mid \mathbf{C}$. So we conduct logistic regression using deep neural networks for predicting the conditional probability distribution of the treatment $W$

---

**Algorithm 1** CBRL.CIV: CIV Regression with Confounding Balancing Representation Learning

---

**Input**: Observational dataset $\mathcal{O} = (W, Y, \mathbf{X})$ with the treatment $W$, the outcome $Y$, the CIV $S$ and the observed variables $\mathbf{C}$; the maximum number of iterations $t$.

**Output**: $\hat{Y}_0 = f_{\gamma^0}(\psi_\theta(\mathbf{C}))$ and $\hat{Y}_1 = f_{\gamma^1}(\psi_\theta(\mathbf{C}))$.

  **CIV Regression**:

  **for** ite=1 to $t$ **do**

    $\{\mathbf{c}_i\}_{i=1}^n \to \varphi_\mu(\mathbf{c}_i) \to P(s_i \mid \mathbf{c}_i)$

    $L_S = \frac{1}{n}\sum_{i=1}^n (s_i log(\varphi_\mu(\mathbf{c}_i)) + (1 - s_i)(1 - log(\varphi_\mu(\mathbf{c}_i))))$

    update $\mu \leftarrow SGD(L_S)$

  **end for**

  **Treatment Regression**:

  **for** ite=1 to $t$ **do**

    $\{\mathbf{c}_i\}_{i=1}^n \to \varphi_\nu(\mathbf{c}_i)$

    $\{\hat{s}_i, \mathbf{c}_i\}_{i=1}^n \to \varphi_\nu(\hat{s}_i, \mathbf{c}_i) \to P(w_i \mid \hat{s}_i, \mathbf{c}_i)$

    $L_W = \frac{1}{n}\sum_{i=1}^n (w_i log(\varphi_\nu(\hat{s}_i, \mathbf{c}_i)) + (1 - w_i)(1 - log(\varphi_\nu(\hat{s}_i, \mathbf{c}_i))))$

    update $\nu \leftarrow SGD(L_W)$

  **end for**

  **Outcome Regression**:

  **for** ite=1 to $t$ **do**

    $\{\mathbf{c}_i\}_{i=1}^n \to \varphi_\mu(\mathbf{c}_i) \to P(s_i \mid \mathbf{c}_i)$

    $\{\hat{s}_i, \mathbf{c}_i\}_{i=1}^n \to \varphi_\nu(\hat{s}_i, \mathbf{c}_i) \to P(w_i \mid \hat{s}_i, \mathbf{c}_i)$

    $\{\psi_\theta(\mathbf{c}_i), \hat{s}_i\}_{i=1}^n \to D(\hat{S}, \psi_\theta(\mathbf{C}))$

    $\{\psi_\theta(\mathbf{c}_i), \hat{w}_i\}_{i=1}^n \to D(\hat{W}, \psi_\theta(\mathbf{C}))$

    $L_Y + \alpha L_{CBW} + \beta L_{CBS} = \frac{1}{n}\sum_{i=1}^n \left( y_i - \sum_{\hat{w}_i \in \{0,1\}} f_{\gamma^{\hat{w}_i}}(\psi_\theta(\mathbf{C}))P(w_i \mid \hat{s}_i, \mathbf{c}_i) \right)^2 +$

    $\alpha D(\hat{W}, \psi_\theta(\mathbf{C})) + \beta D(\hat{S}, \psi_\theta(\mathbf{C}))$

    update $\theta, \gamma^0, \gamma^1 \leftarrow ADAM(L_Y + \alpha L_{CBW} + \beta L_{CBS})$

  **end for**

---

conditioning on $\mathbf{C}$ and $\hat{S}$, namely $P(W \mid \hat{S}, \mathbf{C})$. Next, $\hat{W}$ is sampled from $P(W \mid \hat{S}, \mathbf{C})$ assuming $P(W \mid \hat{S}, \mathbf{C}) = P(W \mid \hat{S}, \mathbf{C}, \mathbf{Z})$ since then through the confounding balance phase, the learned representation $\mathbf{Z}$ and the estimated $\hat{W}$ are independent, i.e., $\hat{W} \perp\!\!\!\perp \mathbf{Z}$ when the optimal solution of Eq. (6) in the main text is obtained. Furthermore, $P(W \mid \hat{S}, \mathbf{C}) \sim P(W \mid \hat{S}, \mathbf{C}) \equiv P(W \mid \hat{S}, \mathbf{C}, \mathbf{U})$ as it is almost surely that $\phi(\mathbf{C}) \equiv \tilde{\phi}(\mathbf{C}, \mathbf{U})$ as stated in Assumption 2. Thus, the causal relationship between $\hat{W}$ and $Y$ is not confounded by $\mathbf{Z}$ and $\mathbf{U}$.

As the defined inverse problem of the standard IV in Eq.(9) in the main text, in our outcome regression stage, $\hat{W} \sim P(W \mid \hat{S}, \mathbf{C})$ and $\mathbf{Z} = \psi_\theta(\mathbf{C})$ are used to predict the counterfactual outcomes. Thus, $\mathbb{E}[Y \mid S, \mathbf{C}] = \int f(W, \mathbf{Z})dP(W \mid \hat{S}, \mathbf{C})$ is the inverse problem for $f(W, \mathbf{Z})$ given $\mathbb{E}[Y \mid S, \mathbf{C}]$ and $P(W \mid \hat{S}, \mathbf{C})$. $\qquad\square$

## C EXPERIMENTS

### C.1 EXPERIMENTAL SETUP

**Implementation sources for comparsion methods.** The implementations for DeepIV, OrthIV, DMLIV, DRIV, SDRIV, and ForestDRIV methods are sourced from EconML (Battocchi et al., 2019). The CBIV implementation is avaiable on the authors' GitHub page at `https://github.com/anpwu/CB-IV`. Similarly, the CFR implementation can be found on the authors' GitHub repository at `https://github.com/clinicalml/cfrnet`.

**Parameters setting.** To ensure a fair comparison, we use the common setting for the tuning parameters as done in Shalit et al. (2017); Wu et al. (2022), such as learning rate and number of

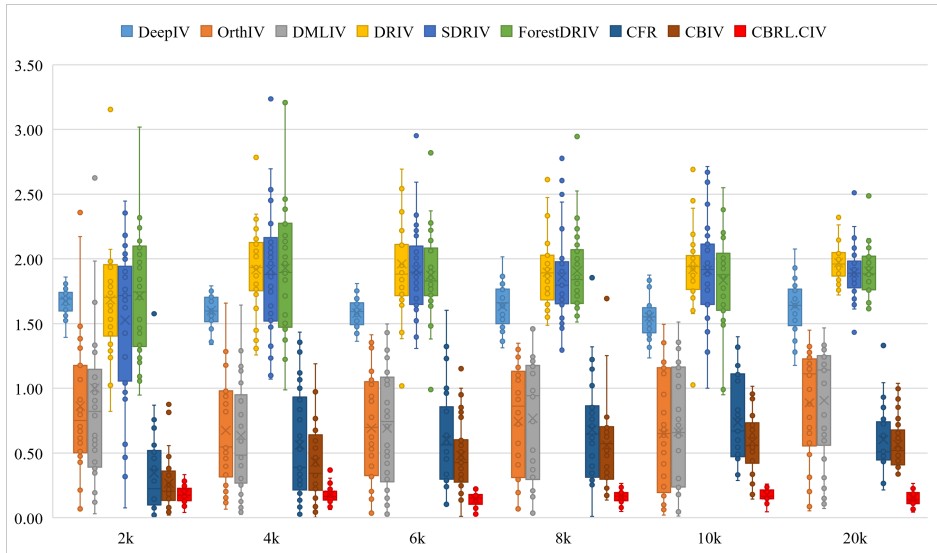

Figure 3: The experimental results of all estimators on within-sample of synthetic datasets with unobserved confounders in non-linear scenarios. The CBRL.CIV obtained the lowest estimation error against all comparisons.

epochs. Furthermore, we set the same parameters, such as the dimension of learned representation **Z** for CFR, CBIV, and our CBRL.CIV.

## C.2 SIMULATION STUDY

We generate the synthetic datasets following the works Hassanpour & Greiner (2020); Wu et al. (2022) for the evaluation. Especially, the set of pre-IV variables **C** and the set of unobserved variables **U** drive from $C_1, C_2, \ldots, C_p, U_1, U_2, \ldots, U_q \sim N(0, \Sigma p + q)$, where $p$ and $q$ are the dimensions of **C** and **U**, respectively, $\Sigma p + q = \mathbf{I}_{p+l} * 0.95 + \varrho_{p+q} * 0.05$ denotes that all values excluding those on the main diagonal are 0.05 in the covariance matrix, and $\varrho_{p+q}$ is a $p + q$ degree all-ones matrix. The CIV $S$, treatment variable $W$, and outcome variable $Y$ are generated as follows:

$$P(S \mid \mathbf{C}) = \frac{1}{1 + exp(-(\sum_{i=1}^{p} C_i + \sum_{i=1}^{q} U_i))}, \qquad (13)$$
$$S \sim Bern(P(S \mid \mathbf{C}))$$

$$P(W \mid S, \mathbf{C}) = \frac{1}{1 + exp(-(S * \sum_{i=1}^{p} C_i + \sum_{i=1}^{p} C_i + \sum_{i=1}^{q} U_i))}, \qquad (14)$$
$$W \sim Bern(P(W \mid S, \mathbf{C}))$$

$$f(W, \mathbf{U}, \mathbf{C}) = \frac{W}{p + q}(\sum_{i=1}^{p} C_i^2 + \sum_{i=1}^{q} U_i^2) + \frac{1 - W}{p + q}(\sum_{i=1}^{p} C_i + \sum_{i=1}^{q} U_i) \qquad (15)$$

where $Bern(P(S \mid \mathbf{C}))$ and $Bern(P(W \mid S, \mathbf{C}))$ are the true CIV and treatment assignment, respectively. In order to avoid bias caused by the data generation process, we repeated the data generation process 30 times (i.e., 30 are generated datasets for each setting) for all experiments. We randomly split a dataset into training (70%) and testing (30%) samples for all our simulations. The generation process is widely used in the evaluation of data-driven causal effect estimation methods Hassanpour & Greiner (2020); Wu et al. (2022); Cheng et al. (2022).

**Results.** The estimation errors of all estimators on within-sample are visualised with boxplots in Fig. 3. Based on Fig. 3, we have the same conclusion drawn in the main text.

Table 3: The table summarises the estimation error of CBRL.CIV with the original $S$ (mean ± STD) over 30 synthetic datasets.

| Syn-4-4 | Syn-8-8 | Syn-12-12 |
|---|---|---|
| 0.48 ± 0.27 | 0.40 ± 0.19 | 0.41 ± 0.14 |

Table 4: The table summarises the estimation error (mean ± STD) over 30 synthetic datasets with different settings and sample sizes. The compared methods have CBRL.CIV, CBRL.CIV with Equation 2 unused (CBRL), and CBRL.CIV with both Equations 2 & 4 unused ($\overline{CBRL}$).

| | | Syn-4-4 | Syn-8-8 | Syn-12-12 |
|---|---|---|---|---|
| 2k | CBRL.CIV | **0.19 ± 1.93** | **0.13 ± 1.95** | **0.09 ± 1.97** |
| | CBRL | 0.26 ± 1.77 | 0.45 ± 1.70 | 0.54 ± 1.72 |
| | $\overline{CBRL}$ | 0.32 ± 1.63 | 0.61 ± 1.68 | 0.76 ± 1.68 |
| 4k | CBRL.CIV | **0.18 ± 0.07** | **0.14 ± 0.03** | **0.13 ± 0.03** |
| | CBRL | 0.43 ± 0.29 | 0.35 ± 0.20 | 0.39 ± 0.11 |
| | $\overline{CBRL}$ | 0.56 ± 0.43 | 0.44 ± 0.32 | 0.61 ± 0.24 |
| 6k | CBRL.CIV | **0.14 ± 0.05** | **0.14 ± 0.03** | **0.12 ± 0.03** |
| | CBRL | 0.48 ± 0.27 | 0.40 ± 0.19 | 0.41 ± 0.14 |
| | $\overline{CBRL}$ | 0.60 ± 0.38 | 0.58 ± 0.31 | 0.51 ± 0.25 |
| 8k | CBRL.CIV | **0.16 ± 0.05** | **0.14 ± 0.03** | **0.11 ± 0.03** |
| | CBRL | 0.56 ± 0.33 | 0.47 ± 0.13 | 0.38 ± 0.12 |
| | $\overline{CBRL}$ | 0.68 ± 0.40 | 0.59 ± 0.25 | 0.47 ± 0.25 |
| 10k | CBRL.CIV | **0.18 ± 0.05** | **0.14 ± 0.04** | **0.11 ± 0.04** |
| | CBRL | 0.59 ± 0.25 | 0.40 ± 0.15 | 0.46 ± 0.14 |
| | $\overline{CBRL}$ | 0.74 ± 0.34 | 0.53 ± 0.24 | 0.60 ± 0.20 |
| 20k | CBRL.CIV | **0.15 ± 0.06** | **0.13 ± 0.04** | **0.11 ± 0.03** |
| | CBRL | 0.57 ± 0.20 | 0.49 ± 0.14 | 0.48 ± 0.08 |
| | $\overline{CBRL}$ | 0.60 ± 0.24 | 0.61 ± 0.28 | 0.58 ± 0.21 |

## C.3 ABLATION STUDY

We have tried using Eq. 2 (i.e., no longer using loss 1) on the synthetic datasets used in Section 4.2. The experimental results on datasets with out-of-samples are reported in Table 3.

**Results.** From Table 3, we know that CBRL.CIV without loss 1 has a larger estimation error than CBRL.CIV with loss 1, i.e., $S \perp \mathbf{Z}$ is not enough for mitigating the confounding bias between $S$ and $W$.

To show the necessity of Eq. 2 (and Eq. 4), we conduct an ablation study and report the experimental results in Table 4. The synthetic dataset generation process is the same as that in Section 4.2.

**Results.** From Table 3, CBRL.CIV has the smallest estimation error on all experimental settings than CBRL (CBRL.CIV without CIV regression, i.e. Eq. 2 is not used) and $\overline{CBRL}$ (CBRL.CIV with both Eqs. 2 & 4 unused). The results indicate that the CIV regression for confounding balance is necessary (and that the treatment regression with confounding balance is also needed).

## C.4 THE EFFECTS OF $\alpha$ AND $\beta$

We have conducted the experiment to examine the effects of $\alpha$ and $\beta$ on the performance of CBRL.CIV. The results are summarised in Table 5. Note that we only adjust the values of $\alpha$ and $\beta$, and Eqs. 1 and 3 were still worked in Eq. 5.

Table 5: The table summarises the estimation error (mean ± STD) over 30 synthetic datasets with different values for $\alpha, \beta$. The sample size is 6k with the setting Syn-4-4.

|  | Within-Sample | Out-of-Sample |
|---|---|---|
| $\alpha, \beta = 0$ | 0.1594 ± 0.0371 | 0.1597 ± 0.037 |
| $\alpha, \beta = 0.01$ | 0.1553 ± 0.0418 | 0.1556 ± 0.0411 |
| $\alpha, \beta = 0.1$ | **0.1406 ± 0.0533** | **0.1412 ± 0.0525** |
| $\alpha, \beta = 1$ | 0.1877 ± 0.0691 | 0.1878 ± 0.0679 |
| $\alpha, \beta = 10$ | 0.2198 ± 0.0511 | 0.2198 ± 0.0511 |

Table 6: The estimation errors (mean±STD) over the within-sample on two real-world datasets with different dimensions of $p$ and $q$.

| Methods | IHDP-2-2 | IHDP-4-4 | IHDP-6-6 | Twins-2-2 | Twins-4-4 | Twins-6-6 |
|---|---|---|---|---|---|---|
| DeepIV | 2.692±0.407 | 2.900±0.290 | 2.516±0.396 | 0.091±0.045 | 0.068±0.048 | 0.082±0.041 |
| OrthIV | 0.773±0.407 | 2.135±0.210 | 3.899±0.735 | 0.256±0.158 | 0.434±0.079 | 0.352±0.113 |
| DMLIV | 1.134±0.881 | 5.010±2.313 | 4.239±1.315 | 0.244±0.156 | 0.425±0.079 | 0.343±0.118 |
| DRIV | 3.872±1.157 | 3.956±0.740 | 4.483±2.550 | 0.285±0.228 | 0.246±0.205 | 0.478±0.329 |
| SDRIV | 8.212±8.106 | 2.932±4.171 | 3.733±4.470 | 0.347±0.242 | 0.286±0.193 | 0.572±0.414 |
| ForestDRIV | 10.176±9.892 | 6.333±7.838 | 4.669±4.17071 | 0.377±0.283 | 0.255±0.228 | 0.490±0.355 |
| CFR | 2.568±1.083 | 1.757±0.459 | 2.917±0.428 | 0.037±0.008 | 0.039±0.021 | 0.040±0.027 |
| CBIV | 2.307±0.975 | 1.344±0.443 | 2.836±0.486 | 0.033±0.011 | 0.034±0.026 | 0.035±0.016 |
| CBRL.CIV | **0.251±0.083** | **0.455±0.068** | **0.120±0.078** | **0.030±0.019** | **0.027±0.018** | **0.031±0.024** |

**Results.** From Table 5, we observe that when $\alpha$ and $\beta$ are set to 0.1, the minimum estimation error is achieved. Furthermore, even when $\alpha$ and $\beta$ are both set to 0, CBRL.CIV still exhibits minimal estimation error.

## C.5 TWO REAL-WORLD DATASETS

**IHDP.** IHDP is the Infant Health and Development Program (IHDP) dataset collected from a randomised controlled experiment that investigated the effect of home visits by health professionals on future cognitive test scores Hill (2011). There are 24 pretreatment variables and 747 infants, including 139 treated (having home visits by health professionals) and 608 controls.

**Twins.** Twins is a benchmark dataset frequently used in causal inference literature. It focuses on twin births and deaths in the USA between 1989 and 1991 Almond et al. (2005). Each twin pair in the dataset includes 40 pretreatment variables that provide information about the parents, pregnancy, and birth of the twins. Both the treated group (W=1, representing the heavier twin) and the control group (W=0, representing the lighter twin) are observed. The true outcome of interest is the mortality rate after one year for each twin.

**Results.** Table 2 shows the estimation errors over the within-sample on two real-world datasets with different dimensions of $p$ and $q$, where $p$ is and $q$ is. From Table 6, we have the same conclusions as described in the main text.

