# OpenReview forum: "Conditional Instrumental Variable Regression with Representation Learning for Causal Inference"
_ICLR.cc/2024/Conference — ICLR 2024 poster_

### Official Review · Reviewer_kNXT · 2023-10-29

**Soundness:** 3 good
**Presentation:** 3 good
**Contribution:** 3 good
**Rating:** 8
**Confidence:** 4

**Summary:**

This paper addresses the challenge of estimating causal effects from observational data while dealing with unobserved confounders. The paper introduces a conditional IV (CIV) approach called CBRL.CIV, which relaxes the strict conditions and allows for non-linearity. Theoretical analysis supports the soundness of CBRL.CIV, and extensive experiments demonstrate its competitive performance in handling non-linear scenarios compared to state-of-the-art IV-based estimators.

**Strengths:**

1. CIV offers a more powerful and flexible approach to handle causal inference with unmeasured confounders.
2. The experimental results are extensive and convincing.

**Weaknesses:**

My main concern is the similarity of this work to Wu.2022 in ICML. Even though the contribution of this work is introducing CIV to handle the restrictive conditions of IV, you methods are similar to Wu et.al. Hence, can you state the difference, or the unique technical contribution of CIV for representation learning?

**Questions:**

See Weaknesses

---

> ### Author Response · Authors · 2023-11-20
>
> We sincerely appreciate your valuable comments and concerns to improve our manuscript and we address them in detail below.
>
> >**Q**. My main concern is the similarity of this work to Wu.2022 in ICML. Even though the contribution of this work is introducing CIV to handle the restrictive conditions of IV, you methods are similar to Wu et.al. Hence, can you state the difference, or the unique technical contribution of CIV for representation learning?
>
> **Response**. Thanks for your question. While our CBRL.CIV method shares a similar idea as CBIV in Wu et al. 2022 [1] in terms of using representation learning in the context of IV-based causal effect estimation, our contribution under the CIV framework is distinct in several key aspects.
>
> (1) Different problem settings as shown in Fig. 1(c) (for CBIV) and Fig. 1(d) (for  our method CBRL.CIV) respectively. Our extends CBIV by providing a more flexible framework that can handle weaker, more realistic conditions. Specifically, CBRL.CIV is designed to work under the setting where $S$ is unconfounded with $W$ and $Y$ conditioning on a set of measured covariates, while CBIV requires the absence of confounding bias between $S$ and $Y$.
>
> (2) The technical contribution of CBRL.CIV lies in its innovative application of confounding balancing to address the confounding bias between $S$ and $W$, especially under the condition of limited sample size. This advancement enables CBRL.CIV to extract a balanced representation $\mathbf{Z}$ for addressing the confounding bias between $S$ and $W$, as well as the confounding bias between $W$ and $Y$. Note that CBIV only mitigates the confounding bias between $W$ and $Y$, but not deal with the confounding bias between $S$ and $W$.
>
> (3) These improvements showcase the practicality and adaptability of CBRL.CIV across a wider range of scenarios in representation learning, where the CBIV method may incur significant estimation bias due to the confounding bias between $S$ and $W$.
> We have revised the following descriptions to make the difference between CBRL.CIV and CBIV clearly in our updated version.
>
> >Current solutions [1,2] consider a standard IV and use a regression model conditioning on $\mathbf{C}$, and they are not applicable to a CIV case since they do not mitigate the confounding bias between $S$ and $W$.
>
> and
>
> >CBIV [2] combines confounding balancing with IV regression to remove the confounding bias between $W$ and $Y$ caused by the unobserved confounders and the observed confounders. Different from these methods, our CBLR.CIV addresses the confounding bias between $S$ and $W$, as well as the confounding bias between $W$ and $Y$. Therefore, CBLR.CIV has a wider application than a method based on a standard IV and it also deals with non-linear cases.
>
> [1] Jason S. Hartford, Greg Lewis, Kevin Leyton-Brown, and Matt Taddy. Deep IV: A flexible approach for counterfactual prediction[C]. In Proceedings of the 34th International Conference on Machine Learning, pp. 1414–1423, 2017.
>
> [2]Anpeng Wu, Kun Kuang, Bo Li, and Fei Wu. Instrumental variable regression with confounder balancing[C]. In International Conference on Machine Learning, pp. 24056–24075. PMLR, 2022.

---

> > ### Comment · Reviewer_kNXT · 2023-12-01
> > **Response to authors**
> >
> > The rebuttal from authors have already solved my main concerns. I would like to raise my score to accept.

---

### Official Review · Reviewer_A6cp · 2023-10-30

**Soundness:** 3 good
**Presentation:** 3 good
**Contribution:** 3 good
**Rating:** 6
**Confidence:** 4

**Summary:**

This paper tackles the complex task of estimating causal effects in observational data while considering unobserved confounders. It introduces a novel approach called CBRL.CIV, which leverages a conditional instrumental variable (CIV) to relax the stringent requirements of standard instrumental variables (IV), such as linearity assumptions and strict unconfounded instrument conditions. Through theoretical analysis and extensive experiments on synthetic and real-world datasets, the paper demonstrates the effectiveness of CBRL.CIV in eliminating confounding bias, particularly in non-linear scenarios, making it a promising alternative to existing IV-based estimators.

**Strengths:**

1. Using CIV to relax the restrictive conditions of IV methods is practical for most realistic scenes.
2. The overall presentation and writing structure is clear and well motivated.
3. Extensive experimental results are convicing.

**Weaknesses:**

1. Using confounder balancing to migrate the gap between treated and controlled groups are very commo. I wonder the specific beneficial point on the combination of CIV and confounder balancing.

**Questions:**

See Weaknesses.

---

> ### Author Response · Authors · 2023-11-20
>
> We are grateful for the constructive suggestions to improve our paper, and we respond to them in detail below.
>
>
> >**Q**. Using confounder balancing to migrate the gap between treated and controlled groups are very common. I wonder the specific beneficial point on the combination of CIV and confounder balancing.
>
> **Response**. Confounding balance is commonly employed to reduce bias in the estimation of causal effects by making the treated and control groups comparable when there are confounders between treatment $W$ and outcome $Y$ and all the confounders are observed. In this work, however, we consider a more complicated problem as depicted in Fig. 1(d), where there exist latent confounders $\mathbf{U}$ affecting $W$ and $Y$ and the distribution of the observed confounders $\mathbf{C}$ is imbalanced not only across the different groups as specified by the different values of $W$ (i.e. the treated and control groups), but also the different groups corresponding to the different values of the CIV $S$. In this case, the usual confounding balancing (on $\mathbf{C}$) cannot be used as some confounders ($\mathbf{U}$) are unobserved, and hence IV is used. Here, as explained in the main text, the standard IV approach is inadequate, thus we use CIV (to address the confounding bias between $W$ and $Y$), and to address the imbalanced distribution of $\mathbf{C}$ across the different groups specified by the different values of CIV $S$ (i.e. the subsequent issue arisen with the use of CIV), we adopt the idea of confounding balancing.
>
> To the best of our knowledge, CBRL.CIV is the first work that considers the imbalance issues in the applications of CIVs when there are finite samples.
>
> We have added some discussion on the specific beneficial points in our updated version.
>
> >To enhance the practical use of the IV approach, specifically to address the double imbalance problem caused by both observed and unobserved confounders in non-linear cases when using a CIV, we propose CBRL.CIV, a novel \underline{C}onfounding \underline{B}alancing \underline{R}epresentation \underline{L}earning method for non-linear \underline{CIV} regression. We further demonstrate its theoretical soundness and practical usefulness.  By integrating CIV with confounding balance, CBRL.CIV emerges as a comprehensive solution that adeptly navigates the intricacies of confounder effects in non-linear models, offering a significant advancement in causal inference.

---

### Official Review · Reviewer_9EAs · 2023-10-31

**Soundness:** 4 excellent
**Presentation:** 3 good
**Contribution:** 3 good
**Rating:** 8
**Confidence:** 4

**Summary:**

This paper presents a novel approach designed to tackle the complexities associated with conditional instrumental variables (CIVs). These are specific variables that, while not qualifying as unconditional instruments, fulfill the IV assumptions when conditioned on a suitable adjustment set.  The paper identifies that the predominant challenge with these CIVs emerges from distributional imbalances among groups delineated by the values in the conditioning set. To mitigate this, the paper proposes a framework that leverages regularization techniques to minimize these distributional discrepancies.

**Strengths:**

1.	The paper is good by its deep exploration of the challenges associated with estimating causal effects in the presence of non-linear relationships and unobserved confounders. It specifically addresses the imbalance issues inherent in applications of CIVs in non-linear settings with finite samples. This is a significant contribution, as CIVs are more practical and less restrictive than traditional IVs in real-world scenarios.
2.	The paper proposes a novel Confounding Balancing Representation Learning (CBRL.CIV) method for non-linear CIV regression in causal effect estimation. Notably, CBRL.CIV efficiently mitigates the confounding biases between $S$ and $W$, as well as between $W$ and $Y$.
3.	The paper delves into the theoretical underpinnings of CBRL.CIV, highlighting its robust capability to generalize across varied groups. This ensures a harmonious balance across the CIV, treatment, and outcome regression networks.
4.	Through experiments on both synthetic and real-world datasets, the effectiveness of CBRL.CIV has been validated in estimating average causal effects. Such empirical evidence is crucial for the adoption and trustworthiness of the proposed method in real-world applications.

**Weaknesses:**

Causal inference fundamentally depends on certain assumptions and the proposed CBRL.CIV method is no exception. If these prerequisites aren't satisfied in specific scenarios, the efficacy of the CBRL.CIV method might be compromised. Thus, a discussion on the potential negative impacts of using CBRL.CIV under such conditions would be beneficial.

**Presentation**:
1.	Figure 2 in the main content and Figure 3 in the appendix should use the same legend.
2.	 When the authors or the publication are included in the sentence, the citation should not be in parenthesis, e.g., “it is well-known that the causal effect of the treatment on the outcome is non-identifiable Pearl (2009); Shpitser & Pearl (2006).” The citation should be in parenthesis using \citep{}.
3.	In Eq. (12), there is a missing square, and the same missing happens in the pseudocode of Algorithm 1.

**Questions:**

Sww weakness

---

> ### Author Response · Authors · 2023-11-20
>
> We sincerely appreciate your constructive comments and suggestions, and we address them in detail below.
>
> >**Q1**. Causal inference fundamentally depends on certain assumptions and the proposed CBRL.CIV method is no exception. If these prerequisites aren't satisfied in specific scenarios, the efficacy of the CBRL.CIV method might be compromised. Thus, a discussion on the potential negative impacts of using CBRL.CIV under such conditions would be beneficial.
>
> **Response1**. Thanks for your suggestions. Following the suggestion, we have included in the revised paper the following discussion on the potential negative impacts associated with employing the CBRL.CIV method as follows.
>
> >The CBRL.CIV method relies on some specific assumptions, such as the existence of a valid CIV and the assumption on independent compliance type (Assumption 2). If these assumptions fail, the method may produce incorrect causal effects. It is vital to verify these conditions and conduct robustness checks and sensitivity analyses to ensure the validity of the causal estimates by CBRL.CIV when the domain experts are not sure whether the conditions are satisfied.
>
> >**Q2**. 1. Figure 2 in the main content and Figure 3 in the appendix should use the same legend.
>
> **Response2**. Thanks. We have used the same legends in the updated version.
>
> >**Q3**. When the authors or the publication are included in the sentence, the citation should not be in parenthesis, e.g., “it is well-known that the causal effect of the treatment on the outcome is non-identifiable Pearl (2009); Shpitser & Pearl (2006).” The citation should be in parenthesis using \citep{}.
>
> **Response3**. Thanks. We have corrected them accordingly.
>
> >**Q4**. In Eq. (12), there is a missing square, and the same missing happens in the pseudocode of Algorithm 1.
>
> **Response4**. Thank you for pointing out these typos. We have corrected them in our updated version.

---

### Official Review · Reviewer_MNhf · 2023-11-05

**Soundness:** 3 good
**Presentation:** 2 fair
**Contribution:** 2 fair
**Rating:** 5
**Confidence:** 3

**Summary:**

This paper introduces a method aimed at addressing the challenge of conditional instrumental variables, which refer to variables that may not be valid unconditional instruments, but satisfy the instrumental variable assumptions when conditioned on an appropriate adjustment set. The paper argues that the primary issue associated with conditional instrumental variables lies in the distributional imbalance observed among the groups defined by the values within the conditioning set. To address this, the paper proposes a methodology that uses regularization similar to the approach employed by Shalit et al. [2017] to minimize these distributional shifts.

**Strengths:**

The paper studies balancing in the conditional instrumental variable setting, and they use an approach reminiscent of Shalit et al [2017] to penalize changes in the conditional distribution of the treatment and instrument are stable across conditions.

I think that it is under appreciated that conditional IVs are possible (although they are implicit in most standard IV assumptions), so I'm glad this topic is getting attention.

I reviewed an earlier version of this paper and most of my critiques from then have now been fixed (with the exception of the experiments below).

**Weaknesses:**

The experiments remain the weakest part of this paper. The result of the proposed method are very impressive relative to the baselines, but it is not at all clear how fair the comparison was. The synthetic data is a novel benchmark, so I would expect that the baselines would all underperform without some hyper parameter tuning, but there is no mention of this (the authors do tune the hyper parameters of the proposed method).

More importantly, both IHDP and Twins are datasets that are used in the unconfounded setting, so it's not clear that they even have an instrument (particularly not a conditional instrument) or unobserved confounding.

Code is supplied but only for the proposed method (not the baselines), and the data generating process is only supplied

**Questions:**

How were the baselines tuned?

Why are they so much higher variance than the proposed method?

If you consider a setting with a standard IV, do the baselines then outperform your method?

How did you select instruments for IHDP and Twins?

What is a plausible source of unobserved confounding for IHDP and Twins?

---

> ### Author Response · Authors · 2023-11-20
>
> We are grateful to Reviewer MNhf for your insightful comments during the last round of our submission. We have revised our manuscript and addressed most of your concerns. We hope the following responses will clarify your concerns about our experiments.
>
> >**Q1**.  How were the baselines tuned?
>
> **Response1**. We use the same setting for tunning the parameters as done in [1], such as learning rate and epoch. To obtain a fair comparison, we set the same parameters for all methods, including our CBRL.CIV.
> We have added the implementations and parameter settings of compared estimators in Appendix C.1.
>
> >**Q2**. Why are they so much higher variance than the proposed method?
>
> **Response2**. These baseline methods are not designed to address the imbalanced distribution. With the data we generated for the experiment, the distribution of $\mathbf{C}$ is imbalanced between different groups specified by $S$ or $W$ due to the influence of latent confounders. When these algorithms are applied to such datasets, the estimation results exhibit significant instability with high variance.
>
> >**Q3**. If you consider a setting with a standard IV, do the baselines then outperform your method?
>
> **Response3**. We assume that the reviewer is asking about the case when there is no confounding bias between $S$ and $W$. In this case, our CBRL.CIV method will become CBIV, i.e., the CIV regression yields $\hat{S}\sim P(S\mid\mathbf{C})=P(S)$ due to $S$ and $\mathbf{C}$ are independent. That is, in this case, CBRL.CIV and CBIV would achieve the same performance in theory, and the experiments in [1] have verified the superiority of  CBIV (and our method) compared to the other baselines.
>
> >**Q4**. How did you select instruments for IHDP and Twins? What is a plausible source of unobserved confounding for IHDP and Twins?
>
> **Response4**. We would like to first clarify that in the Introduction, a real-case scenario is presented to demonstrate the plausibility of CIV and its associated conditioning set. This example also illustrates that latent confounders are commonplace in real applications. The example in the Introduction is provided as follows.
>
> >For example, we aim to estimate the effect of a treatment ($W$) on the recovery from a disease ($Y$). $W$ and $Y$ are affected by unobserved confounders $\mathbf{U}$, such as the socioeconomic status of patients. The variable of clinical practice guidelines ($S$) for treatment decision $W$ is a CIV with the conditioning set ($\mathbf{C}$), which includes factors such as age, health condition, and an individual's propensity to seek medical assistance.
>
> Evaluating data-driven causal effect estimation methods is challenging in real-world settings due to the absence of known ground truths for causal effects, particularly when latent confounders are present. Therefore, to assess the performance of our CBRL.CIV method, we employ both synthetic and semi-synthetic datasets in our experiments.
>
> We also wish to clarify that we **did not** select instruments for IHDP or Twins since there is no knowledge available about the ground truth instruments of the two datasets. To verify the effectiveness of our CBRL.CIV method, following the semi-synthetic data generation process for IHDP and Twins as used in [1] for evaluating CBIV, we use the original covariates to construct the latent variable $\mathbf{U}$ and generate the CIV $S$. During the data generation process, covariates $\mathbf{C}$ are initially used to generate the CIV $S$, and then $S$, $\mathbf{C}$ and $\mathbf{U}$ are used to generate the treatment $W$. It is important to note that $\mathbf{C}$ and $\mathbf{U}$, serving as measured and latent confounders respectively, are chosen from the original covariates that are ancestors of, or direct causes of, the outcome $Y$.
>
> The data generation process is described in the main text of our paper (Section 4.3).
>
> [1] Wu A, Kuang K, Li B, et al. Instrumental variable regression with confounder balancing[C]. International Conference on Machine Learning. PMLR, 2022: 24056-24075.

---

### Author Response · Authors · 2023-11-20
**We thank the reviewers for their thoughtful reviews and comments.**

Dear Reviewers, Area Chairs, Senior Area Chairs, and Program Chairs,

We are grateful for the constructive comments and the time you dedicated to reviewing our manuscript. We have thoroughly considered your feedback and have amended the manuscript accordingly.

We have updated a revised manuscript incorporating the comments from reviewers. All changes to the original manuscript are marked in red for your convenience.

Warm regards,

Authors of Submission3957

---

### Meta-Review · Area_Chair_gKyx · 2023-12-04

**Metareview:**

This paper studies the challenging problem of estimating causal effects from observational data, in the presence of unobserved confounders. The authors use a conditional IV (CIV) to relax the unconfounded instrument condition of standard IV and propose a method named non-linear CIV regression with Confounding Balancing Representation Learning (CBRL.CIV), which jointly eliminates the confounding bias from unobserved confounders and balances the observed confounders, without the linearity assumption.

A reasonable amount of discussions took place between the authors and the reviewers and among the reviewers themselves. In the end, we got four reviews with ratings of 5, 8, 6, and 8 with confidence of 3, 4, 4, and 4 respectively. The reviewers appreciate the novel framework and the comprehensive experiments.

Reviewers proposed issues about experimental results (MNhf), assumptions (9EAs), motivation (A6cp), and contribution (kNXT). Fortunately, the authors have addressed the main issues proposed by the reviewers (9EAs, kNXT).

**Justification For Why Not Higher Score:**

The results are interesting but might not have border impact.

**Justification For Why Not Lower Score:**

Most reviewers believe this work is good enough and the issues proposed by most reviewers have been well-addressed by authors.

---

### Decision · Program_Chairs · 2024-01-16

Accept (poster)